# Metabolic Status Influences Probiotic Efficacy for Depression—PRO-DEMET Randomized Clinical Trial Results

**DOI:** 10.3390/nu16091389

**Published:** 2024-05-03

**Authors:** Oliwia Gawlik-Kotelnicka, Aleksandra Margulska, Kacper Płeska, Anna Skowrońska, Dominik Strzelecki

**Affiliations:** 1Department of Affective and Psychotic Disorders, Medical University of Lodz, Czechoslowacka Street 251, 92-216 Lodz, Poland; anna.zabka@gmail.com (A.S.); dominik.strzelecki@umed.lodz.pl (D.S.); 2Department of Adolescent Psychiatry, Medical University of Lodz, Czechoslowacka Street 8/10, 92-216 Lodz, Poland; aleksandra.margulska@umed.lodz.pl; 3Faculty of Medicine, Medical University of Lodz, Kosciuszki Avenue 4, 90-419 Lodz, Poland; pleskakacper@gmail.com

**Keywords:** depression, abdominal obesity, metabolic syndrome, probiotics, anxiety, stress

## Abstract

Probiotics may represent a safe and easy-to-use treatment option for depression or its metabolic comorbidities. However, it is not known whether metabolic features can influence the efficacy of probiotics treatments for depression. This trial involved a parallel-group, prospective, randomized, double-blind, controlled design. In total, 116 participants with depression received a probiotic preparation containing *Lactobacillus helveticus* Rosell^®^-52 and *Bifidobacterium longum* Rosell^®^-175 or placebo over 60 days. The psychometric data were assessed longitudinally at five time-points. Data for blood pressure, body weight, waist circumference, complete blood count, serum levels of C-reactive protein, cholesterol, triglycerides, and fasting glucose were measured at the beginning of the intervention period. There was no advantage of probiotics usage over placebo in the depression score overall (PRO vs. PLC: F(1.92) = 0.58; *p* = 0.45). However, we found a higher rate of minimum clinically important differences in patients supplemented with probiotics than those allocated to placebo generally (74.5 vs. 53.5%; *X*^2^(1,n = 94) = 4.53; *p* = 0.03; NNT = 4.03), as well as in the antidepressant-treated subgroup. Moreover, we found that the more advanced the pre-intervention metabolic abnormalities (such as overweight, excessive central adipose tissue, and liver steatosis), the lower the improvements in psychometric scores. A higher baseline stress level was correlated with better improvements. The current probiotic formulations may only be used as complementary treatments for depressive disorders. Metabolic abnormalities may require more complex treatments. ClinicalTrials.gov identifier: NCT04756544.

## 1. Introduction

Depression is a common illness that affects 280 million people worldwide, with women experiencing depression at a higher rate than men. It is characterized by the occurrence of lowered mood levels, decreased interest in daily activities, lack of pleasure, loss of energy, and decreased thinking ability [1]. Metabolic syndrome (MetS), according to the definition, consists of such disorders as central obesity, dyslipidemia, insulin resistance, and hypertension [2]. A positive correlation between depression and metabolic syndrome has been repeatedly demonstrated [3]. It is estimated that MetS occurs among more than 30% of people suffering from depression and 12.5–31.4% of the general population [4].

Importantly, both obesity and MetS have been found to be independently associated with depressive symptoms and inflammation. A possible pathophysiological overlap is being considered, with chronic low-grade inflammation and dysbiosis being suggested as possible connecting factors [5].

People who suffer from depression have increased concentrations of inflammatory state markers, such as interleukin 6 (IL-6), tumor necrosis factor-alpha (TNF-α), and other interleukins, while interferon-gamma (IFN-γ) levels are decreased [6]. Moreover, the several pathogenic processes that lead to the development of MetS ultimately result in a pro-inflammatory state, which explains why people with MetS also have elevated levels of inflammatory markers, e.g., TNF-α, C-reactive protein (CRP), and IL-6 [7].

The gut microbiota, which consists of approximately 70% Firmicutes and Bacteroides bacteria, also plays an important role in modulating mental health and central nervous system function [8]. This occurs through the microbiota–gut–brain axis, as a bidirectional communication network between the gut and brain. Moreover, as human and animal studies have shown, the composition of the gut microbiota influences the development of depression and anxiety [9]. In addition, dysbiosis can lead to the development of the components of MetS such as dyslipidemia, obesity, and liver steatosis [10].

The probable common etiopathogenesis of depression and MetS has led to a growing interest in interventions on the gut microbiota, including the use of probiotics and prebiotics as supplements that affect the microbiota–gut–brain axis and reduce the risk of depression [11], as well as metabolic syndrome and its sequelae [12]. The use of probiotics, defined as ”microorganisms that, when administered in adequate amounts, confer a health benefit on the host” [13], may have the effect of reducing the intensity of anxiety symptoms [14]. Moreover, recent studies have reported that probiotics used as complementary treatments lead to better results in the management of depression [15,16]. Importantly, a unique class of probiotics known as psychobiotics may generate or promote the synthesis of neurotransmitters, SCFAs, enteroendocrine hormones, and anti-inflammatory cytokines. Psychobiotics may have a wide range of uses, from improved mood and reduced stress levels to acting as an adjuvant in the therapeutic treatment of several neurodevelopmental and neurodegenerative illnesses [17,18,19,20,21].

Targeted interventions on the microbiota using probiotics among MetS patients have indicated propitious effects on obesity, arterial hypertension, glucose metabolism, and dyslipidemia. Nevertheless, more studies need to be conducted to confirm the positive impact of probiotics on MetS [22].

Moreover, probiotic supplementation may restore the imbalances in some inflammatory biomarkers or alleviate the clinical signs of chronic inflammation [23].

Therefore, it is important to identify conditions (including clinical characteristics) that may be supportive of the curative action of probiotics. For instance, there is little but promising evidence of efficacy of probiotics in reducing the risk of depression or anxiety during the perinatal period [24]. Additionally, probiotics may be beneficial in treating overweight-related cognitive impairment and anxiety [25,26,27,28]. However, little is known about whether probiotic mixtures have more favorable effects on psychometric outcomes in metabolic depression versus depression without metabolic abnormalities [29].

However, the metabolic outcomes with predictive value for the efficacy of probiotics in treating depression are not known. Specifically, it is not known whether central obesity, MetS, or its components may be associated with an improvement in depressive symptoms after microbiota-targeted interventions. Finding such connections may allow personalised treatments to be optimized.

Based on the above, the PRO-DEMET randomized controlled trial protocol was constructed [30]. Then, the pilot study was performed with convincing results regarding the feasibility of a whole-scale study [31]. Importantly, several alterations to the study plan were introduced and explained in the publication of the pilot study results, which are discussed throughout the current manuscript.

The study’s main aim was to assess the efficacy of probiotics towards depressive, anxiety, and stress symptoms in patients with depressive disorders stratified by abdominal obesity or metabolic syndrome comorbidity. The secondary aim was to assess the possible predictive value of chosen lifestyle, clinical, or laboratory parameters for the efficacy of probiotics in treating depression.

Our hypothesis was that probiotic supplementation would decrease the level of depressiveness more effectively in metabolic forms of depressive disorders than in depression without metabolic-associated abnormalities.

This manuscript was planned and prepared according to the CONSORT statement guidelines [32].

## 2. Materials and Methods

The PRO-DEMET trial described herein was designed as a single-center, parallel-group, prospective, randomized, double-blind, placebo-controlled study. It took place at the Medical University of Lodz (Poland) between December 2020 and May 2023. The study timeline has been described previously [30] and is shown in Figure 1.

### 2.1. Participants

Adult outpatients (≥18 years) were randomly assigned (1:1) to probiotic (PRO) or placebo (PLC) groups via computer-generated blocked lists stratified by the presence of MetS according to the International Diabetes Federation (IDF). Unblinding was permissible only if any serious adverse events occurred during the trial. Randomization was performed using a computer-based random number generator (https://www.randomizer.org/, accessed on 10 December 2020) operated by an independent researcher.

The study’s entry population finally consisted of 116 patients recruited in primary care and psychiatric outpatient clinics in central Poland through advertisements in social media and using the snowball method.

Regarding the sample size, it was assumed to be at least 40 subjects per PRO or PLC group [30]. However, more participants were recruited considering the possible attrition rate. Due to significant difficulties in enrolling patients with MetS (as reported in the pilot study [31]), we decided to perform a two-arm study controlling for metabolic abnormalities. Patients with AO constituted about half of the studied population and patients with MetS about one-fourth.

The first primary inclusion criterion was a diagnosis of depressive disorders according to the 11th International Classification of Diseases (ICD-11) (depressive episode, recurrent depression, mixed depressive and anxiety disorder or dysthymia) [33]. The additional inclusion criterion was a Montgomery–Asberg Depression Rating Scale (MADRS) score ≥ 13 based on the clinical utility study by Duarte [34]. The exclusion criteria are listed in Appendix B.

### 2.2. Interventions

At the beginning of the intervention period, the study subjects were requested not to make changes in their routine lifestyle activities over the next 60 days. The PRO group received one capsule containing the probiotic mixture powder in the amount of 3 × 10^9^ colony forming units (CFU) containing *Lactobacillus helveticus* Rosell^®^-52, *Bifidobacterium longum* Rosell^®^-175, and excipients (Sanprobi Stress^®^, Sanprobi Sp. z o. o., Sp. k., Szczecin, Poland; probiotic powder manufacturer—Institute Rosell-Lallemand, Montreal, QC, Canada). The PLC group received the same capsule with only the excipients (Sanprobi Sp. z o. o., Sp. k., Szczecin, Poland).

The optimal composition of the probiotic supplement strains, dosage, and intervention length were selected based on our previous investigation [29].

### 2.3. Outcome Measures

The outcome measures are shown in Table 1, as explained in the protocol [30], the pilot study manuscript [31], and Appendix C.

#### 2.3.1. Questionnaires and Scales

The characteristics of the questionnaires used may be found in the protocol [30].

Study-specific questionnaires were used to assess demographic, lifestyle, and health-related data and to gain information on any adverse events or exclusion criteria emerging during the intervention period.

Validated scales were used to study the patients’ diets (the Food Frequency Questionnaire (FFQ) [35]) and assess their symptom severity (the MADRS [36]; Depression, Anxiety and Stress Scale (DASS) [37]); and quality of Life (QoL; the WHO Quality of Life BREF Instrument [38] scores).

The MADRS scoring instructions applied were as follows: 0 to 6 points—the normal range; 7 to 19 points—mild depression; more than 20 points—at least moderate depression [36].

#### 2.3.2. Biological Material

The fasting venous blood samples were collected by qualified nurses (9 mL) in the morning, between 8:00 and 11:00 a.m., after an overnight rest at the beginning (V1) of the intervention period, and the basic laboratory tests were performed in the Department of Laboratory Diagnostics, Central Teaching Hospital, Medical University of Lodz, Poland.

### 2.4. Patient Involvement

The patients were involved in the choice of outcome measures and decisions related to the management and administration of the trial. We carefully assessed the burden of the trial interventions on the patients. We have started disseminating the main results to the trial participants using dedicated website and e-mail messages.

### 2.5. Data Management

The data were catalogued in compliance with the requirements of findability, accessibility, interoperability, and reusability (FAIR) standards and according to the General Data Protection Regulation (EU) 2016/679.

### 2.6. Ethics

The study was conducted in accordance with the Declaration of Helsinki and approved by the Bioethics Committee of the Medical University of Lodz on 15 December 2020 (reference number RNN/228/20/KE).

### 2.7. Statistical Methods

The statistical procedures were performed with JASP 0.18.1 (accessed via https://jasp-stats.org/download/, accessed on 10 February 2024) and STATISTICA 13.1 (TIBCO Software Inc., Palo Alto, CA, USA). The continuous variables were characterized by means with standard deviations and the categorical variables by the number of observations with the proportion (percentage) of the whole. The normality of distribution of continuous variables was tested with a Shapiro–Wilk test. Accordingly, a U-Mann–Whitney test and Kruskal–Wallis test were used to test inter-group differences. For the Mann–Whitney test, the effect size was given by the rank biserial correlation. The associations between variables were tested using Spearman’s correlation coefficients. A repeated measures ANOVA was used to verify whether there were statistically significant differences between variables over time between the probiotic and placebo groups. A multiple linear regression model and logistic regression analysis were used to evaluate the relationship between various predictor variables and primary and secondary outcomes. The significance level was set at *p* <0.05. As we were facing multiple outcome measures, we chose a single primary outcome measure, as well as using point estimate and effect size measures wherever possible [39,40]. 

## 3. Results and Discussion

### 3.1. Study Flowchart

Figure 2 shows the CONSORT flow diagram of the study participants.

Regarding the tolerability, no serious adverse events were observed. The adverse events included an acute upper airway infection (including COVID-19; n = 8), a urinary tract infection (n = 2), a case of diarrhea (n = 2), headaches (n = 2), exacerbation of an allergic asthma (n = 1), and a mild nettle-rash (n = 1). 

Essentially, the numbers of those who were lost to follow-up or discontinued the intervention were very similar in the PRO and PLC groups.

Four patients were excluded from the analysis, which happened before the unblinding. The reasons for definitely feeling better were given spontaneously by the patients themselves at the beginning of the V2 meeting, and included a regular psychotherapy routine being introduced just after the start of the intervention (not reported previously in the MQ), a national exam being passed to be a specialist in the patient’s occupational field, a successfully finished divorce trial, and the completion of a medical diagnostic process that resulted in a significant improvement in physical health. The above were regarded as major exclusion criteria based on the protocol. However, including all completers did not change the results of the analysis regarding the MADRS scores (see Appendix A). The analyses were performed as per-protocol analyses. All the patients who finished the study were compliant, as assessed form the daily medication log. Importantly, all randomized subjects received the allocated intervention. At the same time, we had a moderately high rate of non-completers (dropouts; eight in the PRO and nine in the PLC group). Thus, an intention-to-treat analysis seemed unjustified [41]. Nonetheless, this attrition rate gave the study internal validity [42].

Finally, we analyzed the MADRS scores from 94 participants (one patient was unable to complete to an in-person or online V2 meeting, meaning the MADRS was impossible to perform), the DASS scores from 82 participants, and the QoL scores from 80 participants (some patients did not give back their self-assessment questionnaires, despite several reminders).

### 3.2. Sample Characteristics

The basic, diet-related, clinical, and laboratory sample characteristics are shown in Table 2. Importantly, the entry PRO and PLC groups did not differ in terms of their sociodemographic, general-health-related, or metabolic-health-associated data. The dietary intakes did not significantly differ between the two groups except for dairy and eggs. Among the psychometric parameters, only the neurovegetative domain of the MADRS was higher in the PRO than the PLC group. The lymphocyte (LYM) level was the only inflammation marker that was lower in the PRO compared with the PLC group. An apparent lack of virtually any differences between the PLC and PRO groups represented an obvious strength of our study.

### 3.3. Changes in Psychometric Data

Table 3 and Table 4 present a summary of the intervention results measured by the psychometric scales.

Significant but similar improvements in MADRS scores were shown in both the PRO and PLC groups after the intervention (PRO vs. PLC: F(1.92) = 0.58; *p* = 0.45). Moreover, there was no difference in delta MADRS scores (ΔMADRS; U = 961; Z = 1.02; *p* = 0.31) nor in percentage delta MADRS scores (%ΔMADRS; U = 1003.5; Z = 88; *p* = 0.38; Figure 3B) between the PRO and PLC groups (Figure 3A).

Consequently, no differences were observed regarding the ΔMADRS domain scores (sadness F(1.73) = 0.42, *p* = 0.52; neurovegetative F(1.73) = 1.20, *p* = 0.28; detachment F(1.73) = 0.56, *p* = 0.46; negative thoughts F(1.73) = 0.25, *p* = 0.62; Figure 3B) nor the %ΔMADRS domain scores between the PRO and PLC groups.

Similarly, the response and remission rates did not differ significantly between the PRO and PLC groups. Interestingly, the subjects supplemented with PRO showed a higher rate of MCIDs (n = 38) as compared with the participants supplemented with PLC (n = 23) (74.5 vs. 53.5%; *X*^2^(1,n = 94) = 4.53; *p* = 0.03; Figure 3C). The effect size, as measured by Cohen’s d score, was d = 0.45, indicating a medium effect, and the number needed to treat was NNT = 4.03. The effect of the PRO remained in a subpopulation treated with antidepressants (n = 66) (75.0 vs. 50.0%; *X*^2^(1,n = 94) = 4.42; *p* = 0.04; d = 0.44; NNT = 4.07) but not in subjects not treated with antidepressants (n = 19) (*p* = 0.51); in a subpopulation treated with selective serotonin reuptake inhibitors (SSRIs), similar findings were shown (70.8 vs. 41.2; *p* = 0.058) (Figure 3D). Importantly, the antidepressant-treated subjects within the PRO group had lower basal DASS and D-DASS scores than the participants not treated with antidepressants (see Appendix A).

The total DASS score changes, as well as the depression, anxiety, and stress domain score changes, were similar in the PRO and PLC groups (F(4.20) = 0.42, *p* = 0.79). Longitudinal data from DASS measurements at five time-points were additionally assessed, stratified by an antidepressant treatment, and no differences were shown between the PRO and PLC groups (Figure 4).

Moreover, there were no differences between the PRO and PLC groups for the QoL score (F(1.78) = 0.01; *p* = 0.93) or the questionnaire psychological domain score.

A recent meta-analysis of human trials using probiotics demonstrated their possible usefulness in depressive outcome measures [43]. Additionally, probiotics were effective for patients with both mild and moderate depression. This fact places probiotics next to nutritional, dietary, and other lifestyle interventions, which may also be effective for mild depressive symptoms [44]. However, our study did not find any significant change in depression scores overall between the probiotics and placebo groups. We only found higher rates of MCIDs in subjects supplemented with probiotics than those under placebo conditions. In most of the research, probiotics were effective in reducing depressive symptoms only as an add-on treatment [43]. We have confirmed those findings, although again in terms of more frequent minimal differences in depression scores after probiotics treatments compared with placebo. Importantly, the present study involved an outpatient clinical population with depression, while most trials before had investigated depressive symptoms in healthy participants or comorbid or secondary depression patients [45]. However, in contrast to most of the meta-analysis findings [43], sex and age did not influence the efficacy of our intervention. Nonetheless, this may have been due to the insufficient sample size of our single trial. Regarding details of the supplementation protocol, the results of an umbrella meta-analysis [45] suggested administering probiotics for depressive symptoms for at least 8 weeks, which was confirmed to have a minimal effect by our study results. However, due to between-study heterogeneity, no firm conclusion could be drawn about the dosages [45], although in our study the dose of 3 × 10^9^ CFU was shown to be possibly enough to obtain the minimum clinically significant effect for depression.

Additionally, the sample size (n = 95) gives our trial better power than all of the previously published randomized clinical trials performed in clinical populations. Moreover, our two-strain probiotic composition confirmed the possible utility of the *Lactobacillus* spp. and *Bifidobacterium* spp. combination in clinical populations [43]. In detail, we added data regarding the action of specific probiotic strains (*Lactobacillus helveticus* Rosell^®^-52 and *Bifidobacterium longum* Rosell^®^-175) towards negative emotional states. In agreement with our results, no significant difference was found between probiotic and placebo groups in any psychological outcome measures in participants with low mood levels who were not currently taking psychotropic medications [46]. In the general population, however, one study found decreases in somatization, depression, and anger–hostility scores [47], although another study revealed no effects of this intervention on wellbeing, quality of life, emotional regulation, anxiety, mindfulness, and interoceptive awareness [48]. Interestingly, altered brain activity was observed in regions implicated in emotional, cognitive, and face processes in healthy volunteers [49]. Similarly to our study results, the probiotic formulation was shown to be minimally effective as an add-on treatment for depressive symptoms in the clinical population; interestingly, the improvement was correlated with the increases in the levels of brain-derived neurotrophic factor and the tryptophan/isoleucine ratio [50,51]. Nevertheless, the current findings are contrary to those of previous trials indicating an overall significant improvement in depressive symptoms in subjects with subthreshold to moderate depression as a monotherapy; however, the latter intervention combined the probiotic strains and S-adenosyl methionine and was implemented for a period of three months [52].

### 3.4. Pre-Treatment Determinants of Probiotic Efficacy towards Depression

Regarding the hypothesis of the study, there was no difference in ∆MADRS scores between the PRO and PLC groups if stratified by the MetS (*p* = 0.65), HSI > 36 (*p* = 0.95), or abdominal obesity (*p* = 0.67) rates. Moreover, the ∆MADRS scores did not differ between the PRO and PLC groups when stratified by CLGI presence, sex, antidepressant treatment, specific psychiatric diagnosis, or comorbidities. Additionally, no regression model using CLGI presence, sex, antidepressant treatment, specific psychiatric diagnosis, or comorbidities as variables could explain the ∆MADRS scores.

The response, MCID, CMC, and remission rates were not predicted by age, sex, MetS, or abdominal obesity presence in the logistic regression models.

The frequency rates of MCID did not differ between the PRO and PLC groups when stratified by the presence of MetS or CLGI, or the lipid, glycemic, or BP criteria of MetS.

As such, the participants in the PRO group who had achieved an MCID or CMC were compared.

The MCID achievers in the PRO group (n = 38) were not significantly different from the non-achievers (n = 13). However, a trend toward statistical significance was shown for higher consumption rates by achievers compared to non-achievers of unprocessed meat (2.26 ± 0.5 vs. 1.90 ± 0.6; *p* = 0.051), fish (2.43 ± 0.7 vs. 1.92 ± 0.8; *p* = 0.071), and drinks (2.08 ± 0.5 vs. 1.80 ± 0.3; *p* = 0.054).

It was found that in the PRO but not the PLC group, the CMC achievers (n = 36; 21 in the PRO and 15 in the PLC group) compared with the non-achievers (n = 58; 29 in the PRO and 29 in the PLC group) had lower pre-treatment BMI scores (23.17 ± 5.1 vs. 25.07 ± 3.1; *p* = 0.02), lower HSI scores (31.48 ± 7.0 vs. 33.90 ± 5.6; *p* = 0.04), higher MADRS scores (24.29 ± 6.3 vs. 18.60 ± 4.7; *p* < 0.001), lower QoL scores (69.86 ± 11.5 vs. 77.97 ± 12.9; *p* = 0.02), and lower QoL psychological scores (14.05 ± 3.7 vs. 16.76 ± 3.5; *p* = 0.02).

In concordance with the above findings, interesting correlations were found between the %ΔMADRS, %ΔDASS, and %ΔQoL scores and some of the psychometric, metabolic, and inflammatory data in the PRO group but not in the PLC group (Table 5). Essentially, the metabolic, psychometric, and inflammatory findings were in the vast majority not significantly correlated; specifically, the baseline BMI or HSI scores did not correlate with the baseline MADRS or QoL scores (see Appendix A).

Interestingly, similar associations were shown for the efficacy of antidepressants; higher immunometabolic depression index scores, including BMI scores, predicted smaller reductions in depressive symptoms after antidepressant usage but with small effect sizes and inconsistent associations [53]. As antidepressants may act as modulators of gut microbiota, the underlying mechanisms of the described phenomena may share a common part [54].

Moreover, a more severe basal A-DASS score was shown to be connected to better improvements in DASS (U = 42.00; Z = 2.12; *p* = 0.03) and A-DASS (U = 28.50; Z = 2.88; *p* < 0.01) scores in the PRO but not PLC group.

We found that the more advanced the metabolic abnormalities (such as overweight, excessive central adipose tissue, and liver steatosis), the less evident the improvements in the psychometric parameters in a self-assessment scale. We hypothesize that more severe or functionally different forms of dysbiosis connected with higher rates of central fat storage [10] require different or more advanced interventions. These may include multi-strain probiotic formulations, longer durations of supplementation, or different strains of probiotics. On the other hand, an individual’s stress level could influence the self-assessment scale results, as it was significantly correlated with the DASS dimensions but not most of the MADRS domains (see Appendix A).

The above findings are, as far as we know, new to the scientific world, as to the best of our knowledge metabolic parameters have not been assessed as determinants of the efficacy of probiotics for depression so far. Few studies have assessed the efficacy of probiotics in obese patients with depression, and the results are promising for depression but inconclusive for obesity [55]. Nonetheless, none of the trials compared the psychopathology outcomes of interventions between obese and lean subjects. We have not found research on the use of probiotics for depression in patients with comorbid liver steatosis.

Additionally, we found that the stress dimension of psychopathology was the most positively associated with the efficacy of probiotics for self-assessed anxiety, stress, and QoL improvements. Importantly, having correlated psychopathological pre-intervention data, we have found that the S-DASS was the only outlier (see Appendix A). In line with this finding, another study found that the use of probiotics could reduce the subjective stress levels of healthy participants and improve their stress-related subclinical anxiety or depression symptoms [56]. Additionally, strain-dependent effects on outcomes related specifically to stress were found in animal studies [57].

Furthermore, a higher LYM level was positively correlated with an improvement in QoL after treatment with probiotics. A lower LYM level may be a result of chronic stress of a different origin (hypercortisolemia) [58], and the data on the actions of probiotics in different baseline cortisol conditions remains inconclusive [56,59], with the topic requiring further investigation.

As the majority of the gut microbiota is thought to be influenced by diet [60], we hypothesized that the diet’s composition would influence the efficacy of the probiotics. Surprisingly, it was shown to be non-significant. The findings may be explained by the fact that we assessed only dietary habits and not anti-inflammatory or microbiota-affecting indices. However, a trend toward significance was shown for higher consumption rates of unprocessed meat, fish, and drinks, including juices, in participants who had achieved an MCID. In line with this finding, a recent meta-review supported the evidence for the relevance of diet and other lifestyle habits in psychiatric treatments [61]. This may be due to anti-inflammatory, antioxidative, or microbiota-modulating actions [62]. The physical activity level was also shown to be non-significant. Nonetheless, it was the only index with a large amount of missing data. Overall, significant interactions between healthy behaviours and probiotic positive effects on anxiety and emotional regulation were shown by another study [48].

Romijn et al. revealed that the baseline vitamin D level influenced the treatment effect of probiotics [46]. We did not measure the level of vitamin D; however, we gathered information on vitamin D supplementation. There was no influence of this supplementation on the intervention outcome measures in our trial.

Mood disorders were shown to be connected with increased gut permeability [63]. In our previous study, a statistically significant positive correlation between I-FABP and anxiety levels was found (in review). However, in the present study, I-FABP was not connected with the efficacy of probiotics for any of the dimensions of negative affective states. This may have been because I-FABP is a marker of increased intestinal permeability only when enterocyte microdamage occurs [63].

Finally, our findings have indicated that probiotic supplementation is safe and well-tolerated.

## 4. Strengths and Limitations

We used a diverse range of outcome measures and both professional-assessed and self-assessed psychometric scales. Some of the outcomes were new in the field, e.g., the WWI, WHtR, and inflammatory markers calculated from CBC findings, such as the MON/LYM ratio. Moreover, we used quite restrictive exclusion criteria and controlled for known confounding factors affecting microbiota, e.g., diet or physical activity.

Our study had several limitations. The sample size was small or modest; however, to the best of our knowledge, none of the previously published trials on probiotics in depression exceeded that number of participants. We did not confirm diagnoses of fatty liver nor measured percentages of body fat. We did not obtain data on gastrointestinal symptoms either. More advanced indicators of inflammation or dysbiosis should probably have been used, such as Il-6 or the gut microbiota composition. Further, we possibly should not have excluded all of the confounders, e.g., unrecognized chronic inflammatory diseases, hormonal contraceptive use, or menstrual phase.

Finally, it is worth noting that the variance in intervention outcomes may be explained by non-specific factors. The significant expectancy effect, with a large effect size, may have played a huge role in the current study.

## 5. Conclusions

To conclude, currently probiotics formulations may only be used as a complementary treatment for depressive disorders. Importantly, comorbid obesity or liver steatosis may influence the efficacy of probiotics treatments for depression, anxiety, and stress. However, further research on the details of such interventions is essential.

## Figures and Tables

**Figure 1 nutrients-16-01389-f001:**
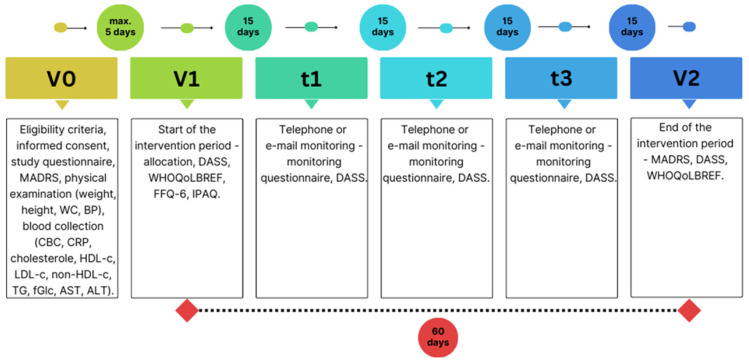
The study timeline. Abbreviations: MADRS—Montgomery–Asberg Depression Rating Scale; WC—waist circumference; BP—blood pressure; CBC—complete blood count; CRP—C-reactive protein; HDL-c—high-density lipoprotein cholesterol; LDL-c—low-density lipoprotein cholesterol; fGlc—fasting glucose; TG—triglycerides; ALT—alanine aminotransferase; AST—aspartate aminotransferase; DASS—Depression, Anxiety, Stress Scale; WHOQoLBREF—WHO Quality of Life BREF Instrument; FFQ—Food Frequency Questionnaire; IPAQ—International Physical Activity Questionnaire.

**Figure 2 nutrients-16-01389-f002:**
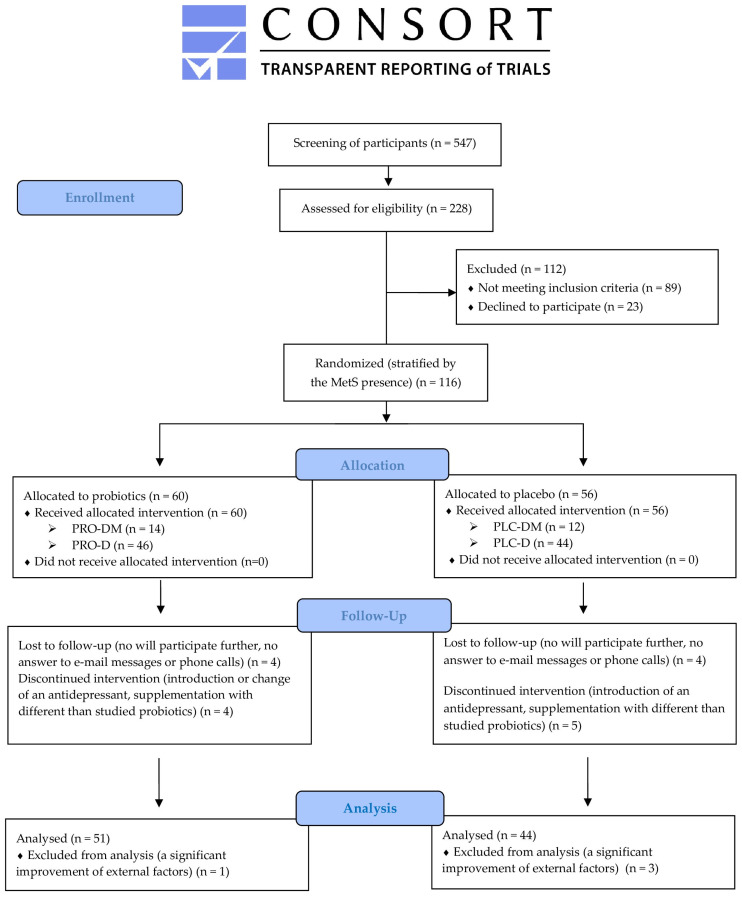
Participant flow diagram. MetS: metabolic syndrome; PRO-DM: probiotic + depression + MetS group; PRO-D: probiotic + depression group; PLC-DM: placebo + depression + MetS group; PLC-D: placebo + depression group.

**Figure 3 nutrients-16-01389-f003:**
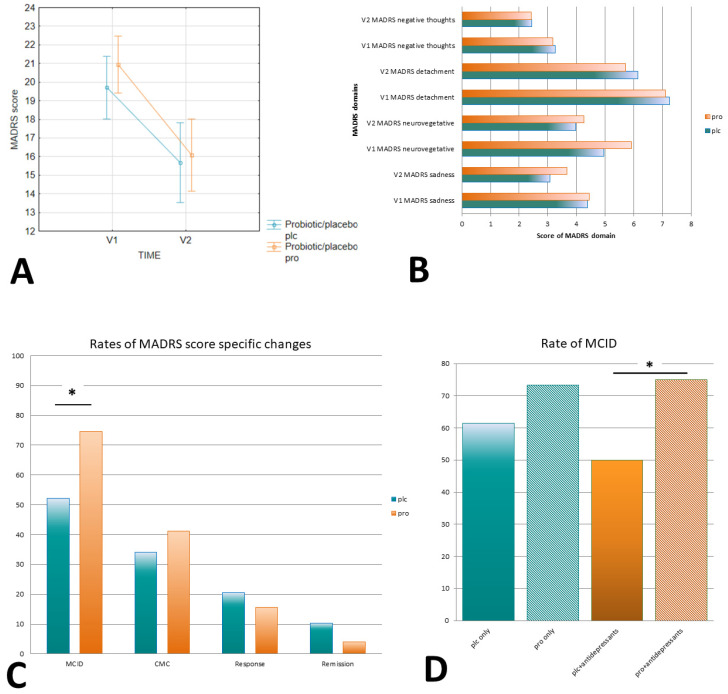
The influence of probiotic supplementation on MADRS parameter score changes. (**A**) %ΔMADRS distribution; (**B**) MADRS domain scores at V1 and V2 time-points; (**C**) rates of MADRS score-specific changes; (**D**) rates of MCIDs depending on antidepressant treatment. Note: * *p* < 0.05. Abbreviations: PRO—probiotic; PLC—placebo; MADRS—Montgomery–Asberg Depression Rating Scale; V1—the start of the intervention; V2—the end of the intervention; MCID—minimum clinically significant difference; CMC—clinically meaningful change.

**Figure 4 nutrients-16-01389-f004:**
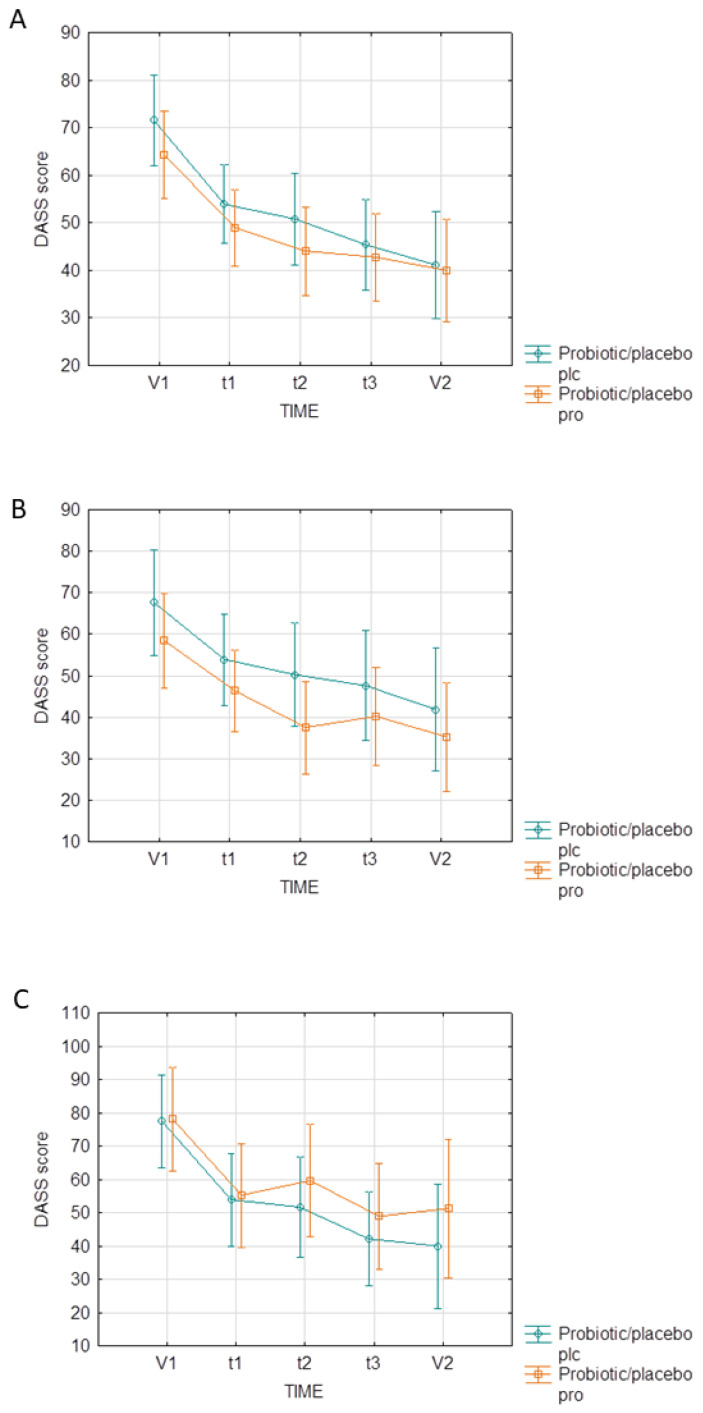
The influence of probiotic supplementation on assessments of the DASS score at five time-points: (**A**) total sample; (**B**) antidepressant-treated subjects; (**C**) subjects not treated with antidepressants. Abbreviations: DASS—Depression, Anxiety, Stress Scale; V1—the beginning of the intervention (0 days); V2—the end of the intervention (60 days); t1—the first monitoring point (15 days); t2—the second monitoring point (30 days); t3—the third monitoring point (45 days).

**Table 1 nutrients-16-01389-t001:** The PRO-DEMET clinical trial outcome measures.

Outcome Measures	
Primary	ΔMADRS
Secondary	%ΔMADRS, MCID MADRS, CMC MADRS, response MADRS, remission MADRS, ΔDASS, %ΔDASS, ΔD-DASS, %ΔD-DASS, ΔA-DASS, %ΔA-DASS, ΔS-DASS, %ΔS-DASS, MCID DASS, MCID D-DASS, MCID A-DASS, MCID S-DASS, ΔQoL, %ΔQoL, ΔQoLpsy, %ΔQoLpsy
Tertiary (baseline only)	AO presence, MetS presence, weight, WC, WWI, BP, fGlc, HDL-c, non-HDL-c, TG, ALT, AST, TG/HDL-c, ALT/AST, HSI, MADRS, DASS, QoL, CLGI presence, CRP, NEU, LYM, MON, PLT, NEU/LYM, MON/LYM, PLT/LYM, SII, I-FABP, dietary habits, physical activity level, antidepressant treatment

Abbreviations: MCID—minimum clinically important difference; MADRS—Montgomery–Asberg Depression Rating Scale; Δ—change between the end (V2) and the beginning (V1) of the intervention period; %Δ—percentage Δ; CMC—clinically meaningful change; DASS—Depression, Anxiety, and Stress Scale; D-DASS—Depression–DASS; A-DASS—Anxiety–DASS; S-DASS—Stress–DASS; QoL—quality of life; QoLpsy—psychological QoL; AO—abdominal obesity; MetS—metabolic syndrome; WC—waist circumference; WWI—Weight-Adjusted Waist Index; BP—blood pressure; fGlc—fasting glucose; HDL-c—high-density lipoprotein cholesterol; TG—triglycerides; ALT—alanine aminotransferase; AST—aspartate aminotransferase; HSI—Hepatic Steatosis Index; CLGI—chronic low-grade inflammation; CRP—C-reactive protein; NEU—neutrophils; LYM—lymphocytes; MON—monocytes; PLT—platelets; SII—Systemic Inflammatory Index, I-FABP—intestinal fatty-acid-binding protein.

**Table 2 nutrients-16-01389-t002:** Study population characteristics. Data are shown as n (%) or the mean ± standard deviation.

	Total (n = 95)	Probiotic (n = 51)	Placebo (n = 44)	*p*	Missing Data (%)
Basic characteristics					
Sex (F:M)	81:14 (85.3:14.7%)	43:8 (84.3:15.7%)	38:6 (86.4:13.6%)	0.78	0
Age (y)	34.4 ± 13.5	34.1 ± 12.2	34.6 ± 14.7	0.75	0
Ethnicity (%) Caucasian	95 (100%)	51 (100%)	44 (100%)		0
Diagnosis according to ICD-11 (6A70:6A71:6A73)	8:26:61 (8.4:27.4:64.2%)	7:16:28 (13.7:31.4:54.9%)	1:10:33 (2.3:22.7:75%)	0.06	0
Psychotropic medications	66 (69.5%)	36 (70.6%)	30 (68.2%)	0.80	0
Antidepressants	66 (69.5%)	36 (70.6%)	30 (68.2%)	0.80	0
Antipsychotics	4 (4.2%)	3 (5.9%)	1 (2.3%)	0.38	0
Comorbidities	51 (53.7%)	29 (56.9%)	22 (50.0%)	0.50	0
AO IDF Polish 2022 guidelines	54 (56.8%) 34 (35.8%)	28 (54.9%) 17 (33.3%)	26 (59.1%) 17 (38.6%)	0.68 0.59	0
MetS IDF Polish 2022 guidelines	23 (24.2%) 24 (25.3%)	11 (21.6%) 12 (23.5%)	12 (27.3%) 12 (27.3%)	0.78 0.68	0
Different than psychotropics pharmacological treatment	33 (34.7%)	15 (27.3%)	18 (41.2%)	0.16	0
Smoking cigarettes	14 (14.7%)	9 (18.2%)	5 (11.8%)	0.38	0
Dietary supplements	49 (51.6%)	24 (45.5%)	25 (56.9%)	0.27	0
Physical activity [MET-min/week]	1968.91 ± 1401.6	2056.72 ± 1578.0	1882.10 ± 1236.6	0.84	58
Dietary habits Food frequency intake assessed on a scale of 1–6: 1—never or almost never; 2—once a month; 3—several times a month; 4—several times a week; 5—every day; 6—several times a day.					
Sweets and snacks	2.63 ± 0.7	2.54 ± 0.7	2.74 ± 0.7	0.28	2.1
Diary and eggs	3.08 ± 0.8	3.96 ± 0.7	3.23 ± 0.8	0.04 *	
Cereal products	3.07 ± 0.6	3.03 ± 0.5	3.11 ± 0.7	0.40	
Oils	2.64 ± 0.6	2.56 ± 0.6	2.73 ± 0.7	0.15	
Fruits	2.77 ± 0.5	2.72 ± 0.5	2.83 ± 0.6	0.40	
Vegetables and seeds	3.36 ± 0.6	3.25 ± 0.5	3.48 ± 0.7	0.10	
Meat (including fish)	2.31 ± 0.7	2.31 ± 0.6	2.31 ± 0.9	0.52	
Drinks (excluding water)	2.02 ± 0.6	2.01 ± 0.5	2.04 ± 0.6	0.94	
Processed food products	2.40 ± 0.5	2.34 ± 0.4	2.46 ± 0.5	0.21	
Psychometric data					
MADRS score total	20.43 ± 5.5	20.94 ± 6.1	19.84 ± 4.7	0.47	0
MADRS score domains Sadness Neurovegetative Detachment Negative thoughts	4.42 ± 1.7 5.46 ± 2.1 7.18 ± 2.2 3.21 ± 1.4	4.45 ± 1.7 5.91 ± 2.3 7.11 ± 2.4 3.17 ± 1.4	4.40 ± 1.8 4.96 ± 1.8 7.26 ± 2.0 3.26 ± 1.4	0.97 0.02 * 0.98 0.54	5.3
MADRS score severity Mild depression Moderate depression	48 (50.5%) 47 (49.5%)	24 (47.1%) 27 (52.9%)	24 (54.5%) 20 (45.5%)	0.51 0.34	0
DASS score	64.74 ± 22.6	63.60 ± 22.2	66.07 ± 22.0	0.64	2.1
Depression	21.55 ± 9.7	20.74 ± 10.5	22.49 ± 8.9	0.44	
Anxiety	17.78 ± 8.8	17.84 ± 9.0	17.72 ± 8.2	0.98	
Stress	25.41 ± 9.2	25.02 ± 8.4	25.86 ± 9.4	0.63	
QoL score	73.49 ± 12.3	74.56 ± 12.9	72.26 ± 11.6	0.38	
Physical	18.84 ± 3.9	18.62 ± 4.0	19.09 ± 4.0	0.43	
Psychological	15.41 ± 3.6	15.62 ± 3.8	15.16 ± 3.4	0.63	
Social	8.53 ± 2.4	8.84 ± 2.4	8.16 ± 2.6	0.24	
Environmental	25.25 ± 4.6	25.92 ± 4.9	24.47 ± 4.0	0.12	
Metabolic-health-associated data					
Weight (kg)	70.66 ± 15.7	69.35 ± 15.3	72.17 ± 16.2	0.40	0
BMI (kg/m^2^)	24.88 ± 4.8	24.29 ± 4.1	25.57 ± 5.4	0.33	
WC (cm)	85.27 ± 13.4	84.95 ± 12.1	86.80 ± 14.9	0.41	
WWI (cm/√kg)	10.17 ± 0.8	10.12 ± 0.8	10.23 ± 0.9	0.59	
WHtR (cm/cm)	0.51 ± 0.1	0.49 ± 0.1	0.52 ± 0.1	0.41	
sBP (mmHg)	121.71 ± 14.1	121.90 ± 13.9	121.48 ± 14.4	0.76	
dBP (mmHg)	82.75 ± 8.5	82.88 ± 8.9	82.59 ± 8.1	0.91	
fGlc (mmol/L)	5.20 ± 0.5	5.17 ± 0.5	5.24 ± 0.6	0.50	
HDL-c (mmol/L)	1.65 ± 0.3	1.71 ± 0.4	1.58 ± 0.3	0.053	
non-HDL-c (mmol/L)	3.71 ± 1.1	3.76 ± 1.1	3.66 ± 1.1	0.62	
TG (mmol/L)	1.16 ± 0.7	1.14 ± 0.7	1.18 ± 0.6	0.76	
TG/HDL-c	0.77 ± 0.5	0.73 ± 0.5	0.81 ± 0.5	0.40	
ALT (U/L)	21.61 ± 15.2	21.94 ± 14.1	21.24 ± 16.5	0.37	
ALT/AST	0.85 ± 0.3	0.86 ± 0.4	0.83 ± 0.3	0.89	
HSI	33.33 ± 6.4	32.90 ± 6.3	33.82 ± 6.7	0.51	
Inflammatory data					
CRP (mg/L)	2.06 ± 2.1	2.10 ± 2.0	2.01 ± 2.2	0.84	0
CLGI	25 (26.3%)	14 (27.4%)	11 (25%)	0.79	
WBC (* 10^3^/)µL	6.17 ± 1.5	6.05 ± 1.5	6.31 ± 1.5	0.42	
NEU (* 10^3^/)µL	3.42 ± 1.1	3.39 ± 1.2	3.45 ± 1.0	0.62	
MON (* 10^3^/)µL	0.51 ± 0.2	0.53 ± 0.2	0.49 ± 0.1	0.31	
LYM (* 10^3^/)µL	2.01 ± 0.5	1.89 ± 0.5	2.15 ± 0.6	0.02 *	
PLT (* 10^3^/)µL	280.25 ± 55.7	276.69 ± 54.6	284.39 ± 57.3	0.35	
NEU/LYM	1.80 ± 0.8	1.89 ± 0.9	1.68 ± 0.5	0.31	
MON/LYM	0.27 ± 0.1	0.29 ± 0.1	0.24 ± 0.1	0.02 *	
PLT/LYM	147.14 ± 42.8	152.97 ± 41.3	140.38 ± 44.0	0.04 *	
SII	502.89 ± 236.7	523.88 ± 276.4	478.57 ± 180.2	0.64	
Others					
I-FABP (ng/)mL	1989.4 ± 1247.1	2069.2 ± 925.3	1894.8 ± 1551.7	0.07	1.1

Abbreviations: F—females; M—males; y—years; 6A70—depressive episode; 6A71—recurrent depression; 6A73—mixed depressive and anxiety disorder; MetS—metabolic syndrome; IDF—International Diabetes Federation; MET—Metabolic Equivalent of Task; MADRS—Montgomery–Asberg Depression Rating Scale; DASS—Depression, Anxiety, Stress Scale; QoL—quality of life; BMI—Body Mass Index; WC—waist circumference; WWI—Weight-Adjusted Waist Index; WHtR—Waist to Heigh Ratio; sBP—systolic blood pressure; dBP—diastolic blood pressure; fGlc—fasting glucose; HDL-c—HDL cholesterol; TG—triglycerides; HSI—Hepatic Steatosis Index; ALT—alanine aminotransferase; AST—aspartate aminotransferase; CRP—C-reactive protein; CLGI—chronic low-grade inflammation; WBC—White Blood Cells; NEU—neutrophils; MON—monocytes; LYM—lymphocytes; PLT—platelets; SII—Systemic Infalammatory Index; I-FABP—Intestinal Fatty Acid-Binding Protein; * significant difference between groups.

**Table 3 nutrients-16-01389-t003:** Changes in psychometric scale scores between the V2 and V1 time-points. Values show means ± SD.

	V1 PRO (Mean ± SD)	V2 PRO (Mean ± SD)	Δ PRO (Mean [95% CI])	%Δ PRO (% [95% CI])	V1 PLC (Mean ± SD)	V2 PLC (Mean ± SD)	Δ PLC (Mean [95% CI])	%Δ PLC (% [95% CI])	*p* ∆	Difefrence in Δ PRO–PLC (Mean [95%CI])	Effect Size r (Rank Biserial Correlation)
**MADRS score**	21.0 ± 6.1	16.1 ± 6.4	−4.9 [−6.8 to –2.9]	−20.98 [−29.7 to −12.3]	19.8 ± 4.7	15.9 ± 7.8	−3.7 [−6.0 to −1.5]	−18.02 [−29.1 to −6.9]	0.31	−1.12 [−4.03, 1.8]	0.124
** Sadness**	4.45 ± 1.7	3.67 ± 2.1	−0.87 [−1.6 to −0.1]	−9.08 [−28.0 to 9.9]	4.40 ± 1.8	3.08 ± 2.3	−1.26 [−2.1 to −0.4]	−9.52 [−49.9 to 27.9]	0.42	0.35 [−0.74, 1.44]	−0.109
** Neurovegetative**	5.91 ± 2.3	4.26 ± 2.3	−1.62 [−2.4 to −0.8]	−15.22 [−34.3 to 3.9]	4.96 ± 1.8	3.97 ± 3.0	−1.06 [−2.1 to 0.0]	−6.74 [−38.2 to 24.7]	0.19	−0.71 [−2, 0.58]	0.175
** Detachment**	7.11 ± 2.4	5.72 ± 2.6	−1.4 [−2.1 to −0.7]	−8.29 [−36.7 to 20.1]	7.26 ± 2.0	6.14 ± 2.7	−1.09 [−2.0 to −0.2]	−8.55 [−22.3 to 5.2]	0.26	−0.43 [−1.57, 0.71]	0.153
** Negative thoughts**	3.17 ± 1.4	2.42 ± 1.3	−0.82 [−1.3 to −0.3]	−17.71 [−32.1 to −3.3.]	3.26 ± 1.4	2.44 ± 1.4	−0.68 [−1.3 to −0.1]	−4.05 [−27.8 to 19.7]	0.68	−0.20 [−0.97, 0.58]	0.055
**DASS score**	63.6 ± 22.2	42.4 ± 22.4	−19.9 [−27.1 to −12.6]	−25.67 [−40.6 to −10.7]	66.1 ± 22.0	43.2 ± 27.8	−23.1 [−30.5 to −15.6]	−36.53 [−48.0 to −25.1]	0.51	3.17 [−7.11, 13.44]	−0.085
** Depression**	20.7 ± 10.5	13.8 ± 9.9	−6.3 [−9.0 to −3.5]	−20.81 [−47.6 to 6.0]	22.5 ± 8.9	15.3 ± 11.6	−7.6 [−10.7 to −4.6]	−36.65 [−50.1 to −23.2]	0.50	1.39 [−2.59, 5.37]	−0.095
** Anxiety**	17.8 ± 9.0	10.3 ± 7.2	−6.7 [−9.0 to −4.4]	−33.45 [−49.1 to −17.8]	17.7 ± 8.2	10.9 ± 8.2	−6.6 [−8.9 to −4.5]	−40.42 [−52.9 to −27.9]	0.94	−0.04 [−3.22, 3.14]	−0.030
** Stress**	25.0 ± 8.4	18.3 ± 10.1	−6.6 [−9.6 to −3.0]	−19.98 [−34.6 to −5.4]	25.9 ± 9.4	17.0 ± 11.3	−8.7 [−12.0 to −5.5]	−31.50 [−46.5 to −16.5]	0.33	2.45 [−2.16, 7.06]	−0.141
**QoL score**	74.6 ± 12.9	81.5 ± 13.0	7.4 [3.6 to 11.1]	10.90 [5.0 to 16.8]	72.2 ± 11.6	80.2 ± 16.6	7.6 [3.8 to 11.5]	10.77 [5.2 to 16.3]	0.93	−0.25 [−5.53, 5.03]	−0.012
** Psychological**	15.6 ± 3.8	17.0 ± 4.0	1.4 [0.3 to 2.3]	11.50 [4.3 to 18.7]	15.2 ± 3.4	17.0 ± 4.7	1.9 [0.8 to 2.9]	13.29 [5.8 to 20.8]	0.47	−0.57 [−1.98, 0.84]	0.112

Abbreviations: MADRS—Montgomery–Asberg Depression Rating Scale; DASS—Depression, Anxiety, Stress Scale; QoL—quality of life.

**Table 4 nutrients-16-01389-t004:** Different intervention outcomes according to the MADRS and the DASS.

	PRO	PLC	*p*	OR [95%CI]	NNT
**MCID MADRS (%)**	74.5	53.5	0.03	2.26 [1.05, 5.86]	4
**CMC MADRS (%)**	41.2	34.9	0.53	1.18 [0.56, 2.96]	16
**Response MADRS (%)**	15.7	20.9	0.51	0.61 [0.25, 1.98]	−19
**Remission MADRS (%)**	3.9	9.3	0.29	0.37 [0.15, 1.17]	−7

**MCID DASS (%)**	27.3	28.9	0.87	0.85 [0.37, 2.45]	−107
**MCID D-DASS (%)**	22.7	26.3	0.71	0.75 [0.31, 2.29]	−34
**MCID A-DASS (%)**	29.5	26.3	0.74	1.07 [0.46, 3.11]	26
**MCID S-DASS (%)**	34.1	36.8	0.73	0.88 [0.35, 2.23]	−36

Abbreviations: MADRS—Montgomery–Asberg Depression Rating Scale; DASS—Depression, Anxiety, Stress Scale; MCID—Minimum Clinically Important Difference; CMC—Clinically meaningful Change; D-DASS—Depression-DASS, A-DASS—Anxiety-DASS; S-DASS—Stress-DASS.

**Table 5 nutrients-16-01389-t005:** Correlation heat map between percentage changes (%Δ) of psychometric parameters and chosen pre-treatment data in the PRO group.

	%ΔMADRS	%ΔDASS	%ΔD-DASS	%ΔA-DASS	%ΔS-DASS	%ΔQoL	%ΔQoLpsy
BMI							
WC							
ALT							
ALT/AST							
HSI							
LYM							
V1 MADRS							
V1 DASS							
V1 D-DASS							
V1 A-DASS							
V1 S-DASS							
V1 QoL							
V1 OoL psychological							

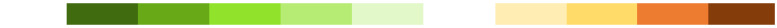
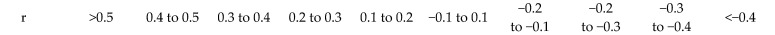

Abbreviations: BMI—Body Mass Index; WC—waist circumference; ALT—alanine aminotransferase; AST—aspartate aminotransferase; HSI—Hepatic Steatosis Index; V1—the start of the intervention period; LYM—lymphocytes; MADRS—Montgomery–Asberg Depression Rating Scale; DASS—Depression, Anxiety, and Stress Scale; QoL—quality of life; r—a correlation coefficient.

## Data Availability

Data will be made available on request.

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
