# Peer review of "Metabolic Status Influences Probiotic Efficacy for Depression—PRO-DEMET Randomized Clinical Trial Results"

_nutrients, 2024, doi:10.3390/nu16091389_

Round 1
Reviewer 1 Report
Comments and Suggestions for Authors
This manuscript reports the results of the PRO-Demet study, which is an interesting trial looking at the effects of a probiotic intervention on depression, according to the metabolic profile of participants. Some of the results are intriguing, however, the manuscript would benefit from editing, as it currently overstates the significance of the findings and several elements of the writing can be improved.
Specific comments:
Abstract:
- The abstract selectively presents only the significant results.
- Review grammar in sentence in lines 24-27 of abstract
Introduction:
- Introduction line 58 – the term ‘Incidence’ is not used appropriately here; also a review paper is cited rather than original data or meta-analyses that show a more direct link to back-up this statement.
- The introduction and discussion are missing some relevant and key recent trials of probiotics as add-on treatment in depression – Scaub et al doi: 10.1038/s41398-022-01977-z. and Nikolova et al. doi:10.1001/jamapsychiatry.2023.1817
- Line 65 – probiotics and prebiotics are not drugs; at most they are LBPs in some instances; replace with supplements or something more appropriate.
- Line 66- use the internationally agreed definition of probiotics instead
- Line 69-70: sentence says nothing at present; please revise or remove.
- Lines 73-75 & Lines 85-86: citing specific original clinical trials would be more relevant than narrative reviews to support these statements.
Methods:
- Sample size calculation: This does not seem to match what’s stated in the protocol paper. Also, why were more participants recruited than the power calc?
- Excluding ppts with spontaneous recovery/recovery possibly not related to the treatment (or what you have called “external factors”) is not common practice in the field, as this will inevitably happen with these conditions; Randomisation is intended to deal with this. Further removal of participants, especially from the placebo group, is likely to skew results in a positive direction. Please justify the choice of approach and reference other major publications that have done this. Alternatively, presenting an ITT analysis with all randomised participants could also be useful.
Results/Discussion:
- Lines 217-218: Justify missing data.
- I would remove the sentence on line 226 “Virtually no missing data is reported” as low rate of missingness is fairly standard in trials.
- MADRS has been analysed not only as %change, but also with all sub-domains, MCID, CMC, response, remission, etc in separate analyses. The primary outcome has not been appropriately defined in the protocol paper nor in the trial registration (it’s simply stated as ‘MADRS’, but not which calculation), therefore this paper’s selection of ‘MCID’ as primary seems outcome-driven.
- Given the multiple analyses conducted on each outcome/set of outcomes I would expect a multiple comparison correction to be applied.
- I am not convinced by the ‘Stress as predictor’ argument – this seems to be based only on several non-parametric correlations with the sub-scale of DASS, which again have not been corrected for multiple comparisons. There are no biological stress markers to back this up and it seems to be just a correlate rather than a predictor.
- For all sub-group analyses mentioned, include the number per group. E.g, antidepressant treated subgroup, CMC achievers in the PRO group, etc.
- Vitamin D is mentioned on p17., but not actually reported anywhere else: “baseline vitamin D level was not found to moderate treatment effect of probiotics in our trial”.
- Line 298: “Nonetheless, in our study the dose of 3×109 CFU was shown to be the minimum efficacious dose” – have you done a dose-finding study? If yes, please reference accordingly; otherwise, revise the statement, as currently you are referencing your earlier narrative review.
- What is the statement on lines 366-367 based on? “We have found that the more advanced metabolic abnormalities (such as overweight, excessive central adipose tissue and liver steatosis), the less evident improvement in psychometric parameters in a self-assessment scale.” Is it based solely on the uncorrected correlations? Doesn’t this contradict the statement of results that: “It was found that in the PRO, but not PLC, group CMC-achievers compared with non-achievers had lower pretreatment BMI (23.17±5.1 vs. 25.07±3.1; p=0.02), lower HSI 340 (31.48±7.0 vs. 33.90±5.6; p=0.04).”
o Limitations & strengths paragraph needs editing as only lines 422-432 have sound arguments.
Conclusion:
- Similarly to the abstract, please be more careful about generalisations and make sure the statements match more closely the findings of the paper.
- The 2nd paragraph can be removed as, as currently written, it doesn’t add anything.
Figures:
- Figure 3a does not present the data in an intuitive/common way; please consider revising the format to be similar to what you have done with DASS.
- Figure 4 – remove the automatically populated annotations by the software above each figure.
- Figure 5 – the colours in the legend do not match the colours in the figure, please correct.
Comments on the Quality of English Language
‘efficient’ used instead of effective or efficacious; ‘insignificant’ used instead of non-significant;
statistical terms such as predictor and moderator used inappropriately in places;
Review grammar throughout
Author Response
Please, see the attachment.

Reviewer 2 Report
Comments and Suggestions for Authors
The studies achieved by the authors had two aims: 1) to assess the efficacy of special probiotics (psychobiotics), on depressive, anxiety, and stress symptoms for patients with depressive disorders and 2) to assess the influence of the value of this chosen lifestyle on the blood biochemical parameters. It is important to mention that more of the patients, during this study, were under treatment with anti-depressive (70%) or antipsychotic (5.9%) medications too
To evaluate the influence of probiotics used, the authors used two methodologies: the survey method, based on periodic questionnaires and periodic biochemical parameter evaluations, performed by blood analysis, performed during 60 days, at the beginning - moment 0 or V1, at the end, (after 60 days) at the moment V2, and at other three moments from this period, noted t1, t2 and t3.
Results obtained during these studies are interesting, showing that under the effect of specific probiotic formulations, an improvement of well-being is possible for these types of patients, and this represents the novelty of this study. This is the reason for which I recommend the publication of this manuscript.
I have two minor recommendations:
1) at the picture 4B, authors must write what represents V1, t1, t2, t3 and V4 ( V1= 0 days; V2= 60 days, t1=...t2=..t3=... because in the manuscript the authors do not specify at which intervals they have done the blood and survey analyses;
2) References appear to not be writted according to MDPI style. Authors must read with attention and writte the references according to these rules, as follow
- Journal references must cite the full title of the paper, page range or article number, and digital object identifier (DOI) where available. Cited journals should be abbreviated according to ISO 4 rules, see the ISSN Center's List of Title Word Abbreviations or CAS's Core Journals List. Note: If you are not sure how to abbreviate a particular journal title, please leave the entire title. The Editorial Office will abbreviate those journal titles appropriately.
- References to books should cite the author(s), title, publisher, publisher location (city and country), publication year, and page:
- In referring to a book written by various contributors, cite author(s) first:
Author Response
Please, see the attachment.
